# Tax abuse—The potential for the Sustainable Development Goals

**Bernadette A. M. O'Hare** [1], **Marisol J. Lopez** [1] *, **Bernadetta Mazimbe** [2], **Stuart Murray** [1], **Nicholas Spencer** [3], **Chris Torrie** [1], **Stephen Hall** [4]

1 School of Medicine, University of St Andrews, St Andrews, United Kingdom, 2 Ministry of Agriculture, Lilongwe, Malawi, 3 Warwick Medical School, University of Warwick, Coventry, United Kingdom, 4 School of Business, University of Leicester, Leicester, United Kingdom

* mjl23@st-andrews.ac.uk

## Abstract

Governments generally provide the services that allow people to access the critical determinants of health: water, sanitation, and education. These are also Sustainable Development Goals and fundamental economic and social human rights. Studies show that governments spend more on public services and health determinants with more revenue. However, governments in low and lower-middle-income countries have small budgets, and tax abuse (avoidance and evasion) contributes to revenue leaks. Researchers have estimated that four countries enable more than half of global tax abuse. We used estimates on tax abuse with a model of the relationship between government revenue and the determinants of health to quantify the potential for progress towards the Sustainable Development Goals 3, 4, 5, and 6. The increase in government revenue equivalent to global tax abuses is associated with 36 million people having access to basic sanitation and 18 million having access to basic drinking water. Additionally, over a ten year period, this increase would be associated with over 600,000 children and almost 80,000 mothers surviving. Thus, curtailing tax abuses would significantly contribute to progress towards the Sustainable Development Goals. Countries that enable tax abuses must review and modify policies to ensure progress towards these goals.

## 1 Introduction and rationale

### 1.1 The determinants of health and the Sustainable Development Goals

Multiple factors (including water, sanitation, and education) provided outside health facilities determine health outcomes. Indeed, half of the reduction in child and maternal mortality between 1990–2015 resulted from increased coverage of these health determinants [1, 2]. Because of their importance to survival, achieving higher coverage of water, sanitation, education, and healthcare are among the Sustainable Development Goals (SDGs), a set of goals agreed by the global community through the United Nations to drive international development and increase global equality [3]. Education, safe drinking water, and safely managed sanitation are SDGs 4 and 6. The SDG 4 and 6 targets are to ensure that all children complete free

**Data Availability Statement:** The data on tax abuse is taken from the State of Tax Justice report and can be accessed here: https://taxjustice.net/

reports/the-state-of-tax-justice-2020/ Our analysis was carried out using the GRADE tool, which can be accessed here: https://med.st-andrews.ac.uk/grade/.

**Funding:** The Scottish Funding Council, the Global Challenges Research Fund, and the Professor Sonia Buist Global Child Health Research Fund provide funding for the GRADE project. The funders had no role in the study design, data collection and analysis, decision to publish, or preparation of the manuscript.

**Competing interests:** The authors declare no conflict of interest.

primary and secondary education and have access to safe drinking water and safely managed sanitation by 2030. Accessing these determinants of health would, in turn, reduce child and maternal mortality rates and contribute to progress towards SDGs 3 and 5 (see S1 Table and S1 Fig) [4]. SDG 3 aims to reduce the under-five mortality (U5M) to less than 25 per 1000 live births and the maternal mortality ratio (MMR) to less than 70 per 100,000 live births in all countries by 2030. SDG 5 aims to increase gender equality, including universal access to reproductive healthcare, which is a significant determinant in the reduction of maternal mortality. Hereafter we refer to these health determinants by the relevant SDG, remembering that achieving these SDGs are a matter of economic and social human rights [5], and currently a substantial number of countries are off track to achieving these goals.

There has been significant progress towards SDG 6, with two billion people gaining access to safely managed drinking water and 2.4 billion to safely managed sanitation between 2000 and 2020. However, many still lack access to water and sanitation. In 2020, two billion people still lacked access to safely managed drinking water and 771 million to basic drinking water, half of which were in Sub-Saharan Africa. In addition, 3.6 billion individuals lacked access to safely managed sanitation, of which 1.7 billion lacked basic sanitation, and almost half a billion practised open defecation. Nine out of ten of those practising open defecation live in Eastern and South-Eastern Asia and Sub-Saharan Africa [6]. These levels of inequality are also apparent when considering progress towards SDG 4. Eighty-eight percent of all children complete secondary education in North America and Europe, compared to only 29% in Sub-Saharan Africa, where only 62% of children complete primary education [6]. In terms of the SDG 3 goals for survival, progress in 53 countries is off-track to achieve the desired child survival rates, of which two-thirds are in Sub-Saharan Africa [6]. In 2017, 86% of the women dying during childbirth from preventable complications were in Sub-Saharan Africa and Asia [6]. Sub-Saharan Africa has the highest rates of maternal and under-5 mortality rates, with 1 in 37 women dying during childbirth in 2020 (compared to 1 in 7800 in Australia and New Zealand) and 1 in 13 children dying before the age of five in 2019 (compared to 1 in 264 in Australia and New Zealand) [6]. Low- and lower-middle-income countries, particularly in Sub-Saharan Africa, are at risk of not reaching the SDGs by 2030, so we must look towards progressive measures to accelerate progress over the next decade and reduce the inequality gap between nations. Increasing government revenue by raising tax revenue could provide the fundamental change necessary to reduce this gap.

## 1.2 Government revenue and the determinants of health

Governments often provide the determinants of health for their citizens as public services and allocate resources to multiple sectors which directly or indirectly impact health. These sectors include education, water and sanitation, healthcare, and these directly affect SDGs 3, 4, 5, and 6 (see S1 Table). Governments in low-income countries have small government revenues ($100 per capita) compared to high-income countries ($14,000 per capita) [7]. However, the impact of an increase in government revenue is more substantial in countries with low revenue per capita, resulting in improved public services and health determinants (see S1 Fig). For example, Reeves et al. showed that an increase in tax revenue of $100 is associated with a $10 increase in government health spending in low- and middle-income countries [8]. Tamarappoo et al. showed that a 10% increase in tax revenue led to a 17% increase in health expenditure in low-income countries [9]. Increased spending results in improved outcomes and economic growth within a relatively short timeframe. Baldacci et al. empirically showed that an increase in government spending by 1% of Gross Domestic Product (GDP) on education is associated with three more years of schooling and a rise of 1·5% growth in per capita GDP. Increasing

health spending by 1% increases child survival by 0·5% and per capita GDP by 0·5%. Health spending benefits are realised immediately, and most of the benefits of spending on education are realised within five years [10]. The relationship between government revenue per capita and child and maternal mortality is non-linear. An increase in revenue has much more impact when the government revenue per capita is small, highlighting that an increase in revenue would have a much more significant effect on low- and lower-middle-income countries. Government revenue also reflects the spending capacity for all sectors, which indirectly affects health, including education and infrastructure [11].

As well as allocating revenue, the critical influence of governance acting via the effectiveness of social spending has been highlighted [12, 13]. Research confirms that good governance amplifies the effect of increased revenue on progress towards the SDGs, especially at low government revenue per capita levels [14]. Corruption is often highlighted as a significant source of revenue leakage within low and lower-middle-income countries. However, a study of 31 African countries showed that a 1% increase in tax to GDP ratio reduces corruption by 0·08 points (measured on a scale of 0–6) [15]. To summarise, an increase in government revenue accelerates progress towards the SDGs and may also strengthen governance, further amplifying this effect. Therefore, it is vital to focus on raising revenue to speed up progress towards the SDGs by considering possible leaks, including tax abuses (see S2 Table), military expenditure, theft from the public purse, and repayment of debts deemed odious [16]. This analysis focuses on tax abuses because individual country estimates on tax evasion and avoidance are newly available [17]. We use these to analyse the potential for progress towards the SDGs critical for health.

## 1.3 Taxation in low- and lower-middle-income countries

The purposes of taxes include revenue generation to pay for public services, the redistribution of wealth, and deter unhealthy behaviours, such as taxes on tobacco [18]. Taxes are a significant form of revenue in low- and lower-middle-income countries, constituting 70% of total revenue [19]. However, government revenue in these countries is low. These countries face enormous challenges broadening their tax base, such as increasing income tax or enlarging the formal sector [20].

Experts believe that increasing revenue from corporate income tax is the most feasible short-term solution to financing public services. Corporate tax contributes much more (about 13%) to low-income countries' tax revenue than high-income countries (about 7%) [21]. Multinational corporations are already operating within the boundaries of low- and lower-middle-income countries, and changes to global tax policy could provide a quick way to increase tax revenues [19]. Therefore, our analysis focuses on the potential for additional revenue from multinational corporations to drive progress towards the SDGs.

Ideally, governments design their tax regimes and expenditure to benefit their citizens. However, they must also shape policies to attract investment as the private sector is the leading provider of employment and capital. As a result, governments compete to attract foreign investment and offer tax incentives, which are not considered tax abuses but reduce tax revenue. Whilst tax incentives have become commonplace for these reasons, there is a debate if they benefit low-income countries. The revenue leaks from incentives and other tax exemptions offered to multinational corporations significantly impact governments' ability to provide essential services and, thus, fulfil human rights obligations. Additionally, governments often compensate for the revenue shortfall resulting from exemptions by raising taxes that impact ordinary citizens, for example, value-added taxes on goods or services [20]. While tax incentives have been empirically linked to corruption and may negatively impact economic

activity [22], governments still offer them out of concern that higher tax rates will deter investors [19]. This competition has been commonly dubbed 'the race to the bottom' and poses a significant threat to sustainable economic development and the progress towards the SDGs.

## 1.4 Tax abuse and government revenue

The international tax framework involves multiple domestic tax codes and bilateral and multilateral tax treaties, resulting in a complex environment for corporations operating in multiple jurisdictions [23]. Sometimes, multinational corporations use artificial tax-avoiding arrangements to move profits from a high to a low tax jurisdiction to minimise the tax due. These arrangements are usually described as tax avoidance and are legal [24]. On the other hand, tax evasion by private individuals or multinational corporations is illegal. However, tax evasion is much smaller relative to tax avoidance, especially in low and lower-middle-income countries, where tax avoidance comes to 5.5.% of tax revenue, and tax evasion comes to 0.3% of tax revenue (see S2 Table). However, as both negatively impact government revenues, public services, and the SDGs, which are fundamental economic and social human rights [25], we follow others and use the umbrella term of 'tax abuse' and hereafter refer to both under this term [26].

After assessing rules and regulations, countries have been ranked by how much they contribute to helping the world's multinational corporations abuse tax by creating vulnerabilities that undermine other countries' revenue. Experts report that upper-middle-income countries and high-income countries enable 98% of global tax abuses. The UK and Overseas Territories and Crown Dependencies, Luxembourg, the Netherlands and Switzerland enable more than half of global tax abuse, with the UK and Overseas Territories and Crown Dependencies enabling 37.4% (see S2 Fig) [17]. Africa and Latin America are most strongly impacted, losing an annual $26 Billion (7% of tax revenue) and $43 Billion (4.2% of tax revenue) to tax abuse, respectively [17].

## 1.5 Aim of the study

This analysis aims to quantify the potential for progress towards the SDGs 3,4,5, and 6 of an increase in government revenue equivalent to the estimated tax abuses in each country where there is data available.

## 2 Methods—data sources

### 2.1 Estimating tax abuses

**2.1.1 Corporate tax avoidance.** Most corporations report the sum of the profits made and costs incurred in different countries as an aggregate of their global operations because they are not required to publicly report this information at the country level. This lack of transparency allows the moving of profits out of the countries with a higher tax rate (often where the actual business occurs) and into countries with low tax rates, thus avoiding paying tax. While many members of the Organisation for Economic Co-operation and Development (OECD) require multinational corporations with revenues over 750 million Euros to report the profit they make and the tax they pay in each country, this is private. However, in 2020, the OECD began to publish their members' data. The report included information on the country where the profit was made, the tax paid, the number of tangible assets and employees, and summarised information such as the number of corporations reporting [27]. However, there are limitations, including a lack of data on the ownership structure of corporations or details on intra-company transactions [28].

Economists in the Tax Justice Network analysed this OECD data to estimate corporate tax avoidance for 215 countries using methodologies previously reported [29–31]. Tax avoidance was assessed using the country-by-country reporting datasets to calculate the mismatch between expected and reported profits, with the former assessed using figures on labour and sales. Profit misalignments were calculated as the difference between expected and reported to provide estimates on the levels of tax avoidance within each country [32].

**2.1.2 Tax evasion.** When corporations and individuals deposit revenues in tax havens, there is an artificial decrease in tax payments and a tax loss in the home country. These types of tax abuses are illegal and much more complex to quantify due to a lack of data. The Tax Justice Network Tax applied an approach based on methodologies by Henry [33], Zucman [34], and Alstadsaeter, Johannesen, and Zucman to estimate the volume of deposits used to evade tax [35]. First, abnormal deposits were identified by pinpointing jurisdictions that had an abnormally high foreign deposit rate relative to the size of the country's economy. Inward bank deposits are highly correlated with GDP (when countries with high levels of bank secrecy are excluded), allowing abnormal deposits to be identified, and traced to their origin countries. Abnormal deposits combined with estimates of total global offshore wealth [36] were used to derive the values of offshore wealth from each country of origin. The losses in income tax revenue were calculated using the tax rate in each origin country [32].

**2.1.3 Conducting secondary analysis on the State of Tax Justice.** We used the figures for total tax abuse in each country (see S2 Table). Multinational corporations shift $1·38 trillion of profit each year into tax havens (see S2 Table). Governments would have taxed a proportion of this, depending on the tax rates in each country. The hiding of profits in tax havens results in countries losing $245 billion corporate tax globally, which is in line with previous estimates, ranging from $90–280 billion [37]. A further $182 billion losses are because of private offshore tax evasion [17]. We conducted secondary data analysis on this report as, firstly, it is the first of its kind to provide a country-by-country breakdown of total tax abuse, and, secondly, the sources and methodologies were described appropriately and transparently to assure their validity and reliability. Additionally, the reliability of secondary data can be verified if similar methods produce similar results [38]. The $427 billion figure annual global tax abuse is similar in scale with figures provided by others ($500–600 billion) [29–31]. The slightly lower estimates indicate that these figures are conservative and that the impact of tax abuse may be more extensive than we can quantify with current data sources.

## 2.2 The GRADE—estimating the impact of an increase in government revenue equivalent to the tax abuse

To estimate the impact of an increase in government revenue equivalent to the tax abuse in each country on progress towards SDGs 3, 4, 5 and 6, (increasing survival, access to basic drinking water/sanitation/education, and increasing gender equality), we employed economic modelling from the Government Revenue and Development Estimations project (the GRADE) [14]. The GRADE uses government revenue data from the UNU Wider database [39] and data on the SDG targets from 217 countries (basic/safe drinking water, basic/safe sanitation, school years and under-5/maternal survival rates) from the World Development Indicators [40] to model the impact of government revenue on progress towards the SDGs. It includes governance indicators from the World Governance Indicators, including political stability and absence of violence, government effectiveness, regulatory quality, the rule of law, and control of corruption (See S3 Table) [41]. The estimates are realistic because it is assumed that governments spend any additional income in the same way they have been in recent years

rather than reallocating funds to a specific sector. Because of this, the GRADE accounts for any change in governance over time [42].

The relationship between government revenue per capita and progress towards the SDGs is non-linear. The best model of this non-linearity is a version of an inverse function that starts with minimal effects for countries with low government revenue per capita; the impact rapidly increases until a saturation point or plateau is reached, and the impact of an increase in revenue on the SDGs slows. Thus countries with small per-capita government revenues have a better scope for progress towards the SDGs with small revenue increases [11]. Rather than impose the same sigmoid shape across all countries, the modelling includes six dimensions of governance from the World Governance Indicators [41], allowing an individual 'sigmoid' 'S' shape, which varies for each country as governance and revenue change. A comparison of the actual and fitted results used by the GRADE modelling confirms how close the modelled estimates are to the existing data on the coverage of the SDG variables, confirming the robustness of the model and the precision of the results presented here [43].

### 2.3 Time selected

For tax avoidance, 2016 figures were used for all countries apart from the US, as a more complete data set for 2017 was available. Figures on tax evasion are from 2018. We converted tax abuse estimates from the year of their origin to 2010 figures as the GRADE uses constant 2010 USD. Tax abuse will fluctuate between years, but as these are the first annual country-by-country estimates, we assumed the losses reported for 2016–2018 occurred every year over the period studied. Most benefits accrue after five years, and the GRADE does not attempt to model or provide estimates for the first five years after the increase in revenue. However, in practice, there would be a gradual accumulation of benefits over the first five years as revenue increases and governments spend money. Therefore, we analysed from 2003, with benefits accrued by 2008 and present these for 2008–2017. We selected 2017 because this is the most recent year with data for most of the required variables. There are slight fluctuations in basic and safely managed water and sanitation coverage due to changes in governance and government revenue per capita. We present the average for the ten years studied for water and sanitation and the total for extra years of education and child and maternal deaths averted.

### 3 Results

This section summarises the potential for progress towards SDGs 3, 4, 5, and 6 if there was an increase in government revenue equivalent to the estimates for tax abuse for the respective countries. However, many countries lack the necessary data for this analysis, meaning that the estimates provided underrepresent the true potential of curtailing these losses. For example, only 36% of countries had data on safe drinking water and only 34% on safe sanitation.

Our country-by-country analyses yielded a significant variation in results based on the deviation in tax abuses and levels of governance within each country. For example, Mozambique had the highest volume of tax abuse within the low-income country bracket with a total of $478 million. We deflated this figure into 2010 US Dollars ($433 Million) as the GRADE modelling uses 2010 constant US Dollars. We estimate that if there were an increase in government revenue equivalent to the tax abuse over ten years, on average, 572,000 people (of which 99,000 are children and 135,000 are women) would have access to basic drinking water, on average 724,000 people (of which 125,000are children and 170,000 are women) would have access to basic sanitation, and 63,000children would attend school for an additional year. These increases would influence the survival rates (see S1 Fig), and 22,879 under-5 and 3,671 maternal deaths would be averted with this increase in revenue. Using this approach for each

                        

country with available data, we summarise the results in S4 Table, which shows the additional progress towards the SDGs associated with increased government revenue equivalent to tax abuse in each of the four income levels.

It is reasonable to assume that the countries and their dependents which create the vulnerabilities that enable 55% of global tax abuse are indirectly responsible for 55% of the reduced coverage of the determinants of health, the associated loss of life, and, thus, progress towards the SDGs, as shown in S5 Table. Accordingly, the UK and its Overseas Territories and Crown Dependencies are indirectly responsible for 37.4% of the coverage decrease in health determinants and the associated loss of life, as shown in S6 Table.

## 4 Discussion

### 4.1 Research in context

Our findings demonstrate that with an increase in revenue equivalent to tax abuse, significant progress could be made towards SDGs 3,4,5, and 6. The increase in government revenue equivalent to global tax abuses is associated with 36 million people accessing basic sanitation, 18 million basic drinking water, and almost 7 million children attending school for an extra year. Additionally, this increase would be associated with over 600,000 children and nearly 80,000 mothers surviving. Our findings support those of other studies, many of which demonstrate that an increase in revenue through curtailing leaks could provide the resources currently missing to ensure all humans can access their right to health. Previous studies have shown that the Millennium Development Goals could have been achieved if losses in government revenue through illicit financial flows, corruption, and debt service were curtailed [44]. Whilst grants are commonly offered to low- and lower-middle-income countries to provide funding for healthcare, these have been shown to be ineffective in providing adequate resources for essential services. Western governments must adopt a no-harm approach by implementing policies to ensure businesses domiciled within their countries do not harm human rights. Curbing losses to tax abuse could provide the necessary revenue to make these changes and ensure development goals are met [42].

Corruption is often cited as a cause of lost revenue. However, the GRADE takes a country's governance into account when modelling the impact of revenue on progress towards the SDGs. As an increase in government revenue has been shown to be associated with an improvement in the quality of governance over time, it is vital to ensure that preventable leaks such as tax abuses are curbed, particularly as a small number of developed countries, such as the UK, contribute to a large proportion of tax abuses. Therefore, focusing on increasing revenue through curtailing leaks to tax abuse will allow countries to develop effective governance and democratic governments that can ensure human rights are fulfilled, and the SDG targets can be met by 2030.

### 4.2 Limitations

Tax abuses are hidden, and indirect methods have evolved to estimate the scale of this concealed phenomenon using limited data. We used the estimates by the State of Tax Justice 2020 because they are available for individual countries and use peer-reviewed methodologies. However, these figures are also limited as they only estimate direct losses through corporate tax avoidance and do not include indirect losses. Indirect losses are three times as large and result from governments trying to counter the direct losses by competing and reducing their effective tax rate. Previous research that estimated both direct and indirect losses have ranged from $500 to $600 billion each year [45], which is, as expected, larger than the figures presented by the Tax Justice Network. It is currently impossible to estimate indirect losses by

individual countries, so the values shown here are conservative. Additionally, the Tax Justice Network figures focus on multinational corporations [17], meaning that tax losses by smaller businesses are not included. On the other hand, this is the amount that could be raised if all tax abuses were curtailed and everything else remains the same, while the actual amount would be determined by responses to the policy measures required to bring this about [46].

Furthermore, tax abuse is a phenomenon that has occurred over many decades, and the estimates provided here are merely a snapshot in time which further results in a conservative estimate. Finally, we have projected the forecast from one year, but tax abuse will fluctuate between years. Still, as these are the first available estimates, and until more estimates become available, our methodology can provide an insight into the scale of the problem and the potential for improvement. A further limitation is the sparse data on the coverage of safely managed water and sanitation services in low- and lower-middle-income countries. This resulted in a substantial underestimate for the potential of curtailing these losses.

## 4.3 The potential for the Sustainable Development Goals of curtailing tax abuses

The median revenue loss from low- and lower-middle-income countries is lower than other income levels, yet the impact on health determinants is much more significant. This difference is because these countries have small revenues per capita. Additional income in such countries would have more impact because the interventions required to reduce child mortality at very high levels are less costly than in wealthier countries. Thus, while low- and middle-income countries lose less in absolute terms than other income groups, they would witness substantial progress towards their SDG targets. In addition, some countries that lose large proportions of their government revenue to tax abuses also lose large proportions of their revenue to external debt service, which further compromises progress towards the SDGs. There is often 100% coverage of basic water and sanitation in upper-middle- and high-income countries. Thus, the higher-income countries' gains are realised mainly in increased numbers accessing safely managed sanitation and additional years of education.

## 4.4 The Sustainable Development Goals, tax abuses and human rights

Achieving the SDGs are a matter of economic and social human rights [5]. Tax abuse undermines government revenue. Therefore, it also undermines access to the determinants of health, which are fundamental human economic and social rights and impede progress towards achieving the SDGs. A country or entity that facilitates these abuses violates their international human rights obligations. The United Nations Economic and Social Council has declared that home countries must prevent their corporations' human rights infringements abroad [47]. The key areas include providing tax havens and remaining passive when multinational corporations engage in transfer pricing to minimise tax paid in other countries (transfer pricing is over-or under-invoicing between different arms of the same companies to minimise paying taxes). The United Nations Committee on the Rights of the Child (UNCRC) has recognised this and recently requested Ireland explain the measures they take to ensure that multinational corporations headquartered in Ireland do not negatively impact children overseas [48].

The countries which contribute most to global tax losses also contribute to overseas development aid. However, it has been shown that a 10% increase in domestic health financing is five times more effective than a 10% increase in development assistance for health in reducing under-five mortality [49]. This finding supports the premise that countries that provide aid should prioritise a review of policies that facilitate tax abuses. Equally, providing technical

assistance for low-income-countries to increase their domestic revenue capacity will likely become an essential component of future development agendas [50].

Host countries of multinational corporations must respect, protect, and fulfil human rights within their territory and jurisdiction. This obligation includes the responsibility to use all available tools at their disposal to protect their citizens and business enterprises against infringements by other actors. Tools include legislation, policies, regulations, and adjudication. In addition, governments must invest in the revenue authorities and review tax incentives and treaties to counter tax abuse and maximise public finances. These actions should be part of every aid-dependent country's strategy to decrease its dependence on aid [20].

## 5 Conclusions and recommendations

Tax abuse contributes to millions not accessing their rights and hinders progress towards the SDGs. The limitations of this study include not having data for abuses by smaller businesses, having estimations for a limited period, the assumption that all other circumstances remain the same, and the lack of data on safe water and sanitation in lower-income countries. These limitations indicate that the figures presented within this paper are an underestimate and that tax abuses have a more extensive impact than can be estimated with current data sources. A significant proportion of global tax abuse has been attributed to four enabling countries: the United Kingdom, the Netherlands, Switzerland, and Luxembourg. The countries creating vulnerabilities to international tax abuses should review their tax policies to ensure they are meeting their human rights obligations and are not negatively impacting government revenue and the progress towards the SDGs. Governments should publicly publish the disaggregated, company-level data to ensure transparency. Curtailing tax abuse would contribute substantially to progress towards the SDGs and result in many accessing their fundamental economic and social rights.

## Supporting information

**S1 Fig. A framework for losses from government revenue and the impact on the Sustainable Development Goals.**
(TIFF)

**S2 Fig. The four countries which enable more than half of global tax abuses [17].**
(TIFF)

**S1 Table. The Sustainable Development Goals (SDGs) with the indicators used in this paper.**
(DOCX)

**S2 Table. Definitions.**
(DOCX)

**S3 Table. World Bank definitions of the determinants of health.**
(DOCX)

**S4 Table. Potential for progress towards the SDGs associated with an increase in government revenue equivalent to the tax abuse for all income levels.**
(DOCX)

**S5 Table. Potential for progress towards the SDGs associated with an increase in government revenue equivalent to the global tax abuse attributable to four enabling countries.**
(DOCX)

**S6 Table. Potential for progress towards the SDGs associated with an increase in government revenue equivalent to the global tax abuse attributable to The UK and Overseas Territories and Crown Dependencies.**
(DOCX)

**S7 Table. Potential for progress towards the SDGs associated with increased government revenue equivalent to the tax abuse in low-income-countries.**
(DOCX)

**S8 Table. Potential for progress towards the SDGs associated with increased government revenue equivalent to the tax abuse for lower-middle-income countries.**
(DOCX)

**S9 Table. Potential for progress towards the SDGs associated with increased government revenue equivalent to the tax abuse for upper-middle-income countries.**
(DOCX)

**S10 Table. Potential for progress towards the SDGs associated with increased government revenue equivalent to the tax abuse for high-income countries.**
(DOCX)

## Acknowledgments

We thank Dr Kyle McNabb for his contribution to our paper through his creation of visual charts on the impact of tax abuse.

## Author Contributions

**Conceptualization:** Bernadette A. M. O'Hare, Nicholas Spencer.

**Data curation:** Bernadetta Mazimbe, Stephen Hall.

**Formal analysis:** Bernadette A. M. O'Hare, Marisol J. Lopez, Bernadetta Mazimbe, Chris Torrie.

**Funding acquisition:** Bernadette A. M. O'Hare, Marisol J. Lopez.

**Investigation:** Marisol J. Lopez.

**Methodology:** Bernadette A. M. O'Hare, Marisol J. Lopez, Stephen Hall.

**Project administration:** Bernadette A. M. O'Hare.

**Resources:** Bernadette A. M. O'Hare.

**Software:** Stuart Murray.

**Supervision:** Bernadette A. M. O'Hare, Stephen Hall.

**Validation:** Bernadette A. M. O'Hare, Marisol J. Lopez, Stuart Murray, Stephen Hall.

**Writing – original draft:** Bernadette A. M. O'Hare, Chris Torrie.

**Writing – review & editing:** Bernadette A. M. O'Hare, Marisol J. Lopez.

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
