## [Decision Letter · Decision Letter 0]

2 Aug 2021

 PGPH-D-21-00278 Tax abuse - the potential for the Sustainable Development Goals PLOS Global Public Health

Dear Dr. Lopez,

Thank you for submitting your manuscript to PLOS Global Public Health. After careful consideration, we feel that it has merit but does not fully meet PLOS Global Public Health’s publication criteria as it currently stands. Therefore, we invite you to submit a revised version of the manuscript that addresses the points raised during the review process.

We look forward to receiving your revised manuscript.

Kind regards,

Augustine D. Asante

Academic Editor

Journal Requirements:

Additional Editor Comments (if provided):

Some additional comments:

1. Authors should provide further details on how tax abuse was quantified, as one of the reviewers pointed out; what they have provided is too sketchy for readers to appreciate the approach/ methodology used.

2. Should explore alternative ways of presenting the results; too many tables are used.

3. Data sources should be indicated under the tables.

4. What sensitivity analysis was conducted to check the robustness of the results?

5. The terms 'tax abuse' and 'tax loss' should be clarified - tax abuse may be a deliberate act to avoid/evade tax which is illegal, tax loss may result from legitimate tax concessions granted to companies by governments.

6. Clarify whether this study is focused on multinational corporations only (one reviewer raised this issue). Note that local companies also evade tax.

7. It would have been helpful to provide some information on corporate tax rates and tax incentives/concessions in LMICs as part of the background material.

Reviewers' comments:

Reviewer's Responses to Questions

**Comments to the Author**

1. Does this manuscript meet PLOS Global Public Health’s publication criteria? Is the manuscript technically sound, and do the data support the conclusions? The manuscript must describe methodologically and ethically rigorous research with conclusions that are appropriately drawn based on the data presented.

Reviewer #1: Yes

Reviewer #2: Partly

2. Has the statistical analysis been performed appropriately and rigorously?

Reviewer #1: Yes

Reviewer #2: No

3. Have the authors made all data underlying the findings in their manuscript fully available (please refer to the Data Availability Statement at the start of the manuscript PDF file)?

Reviewer #1: Yes

Reviewer #2: Yes

4. Is the manuscript presented in an intelligible fashion and written in standard English?

Reviewer #1: Yes

Reviewer #2: No

5. Review Comments to the Author

Reviewer #1: All the additional comments are include in the attachments which I have uploaded.

I suggest that the authors add more information in the Discussion and conclusion.

A papragraph or two about the challenges of corruption in achieving SDGs.

Reviewer #2: Tax abuse – the potential for the sustainable development goals

General

The paper addresses an important issue in public finance with relevance to social sectors including health and education. Tax evasion and avoidance have the potential to transform the economic potentials of several countries, especially in developing regions. However, there are some weaknesses of the paper that needs to be considered in a revised version. I have listed more specific comments below for the consideration of the authors.

Specific comments

1. While the study does well to discuss the pathways in the background, there was little effort to discuss the context of the study. It will be good to have some background statistics to show the extent of the problem, at least across the income levels. For instance, what is the current state of sanitation and water access in the regions? What about tax revenues and evasion? This will help place the study into some better context.

2. The techniques used in this paper are not common. It is therefore important that emphasis is placed on providing sufficient detail to the reader. At the moment this is not the case. The approach to estimate the impact of tax abuse is very nuanced. There should be a systematic discussion on the procedure and approach to arrive at each of the estimates explained in the results.

3. In line with the previous comment, it will be also be helpful to have a clear definition of the variables.

4. Form the explanation provided on tax abuse, it appears the paper only captures tax avoidance and not tax evasion. If this is the cases, kindly clarify. Another concern with the measure of tax abuse is that it appears to only focus on multinationals. This is not quite clear in the paper. If this is the case, it will be good to limit the extent to which results are generalized. It will also be good to acknowledge the limitation.

5. Again, in the estimation of the impact, it is not clear if the tax loss provides for each of the facilities jointly or separately. For instance, in a single country, will that tax loss provide for water, sanitation and schooling jointly? This is not clear. Would be good to discuss these in detail in the methodology.

6. Also, are the deaths averted as a result of the water and sanitation provided?

7. In computing the figures for access to basic facilities, did you use cost of each of the facilities within countries? How exactly were these numbers arrived at?

8. The presentation of the tables can be improved. In the current form, there is so much information in each table. Also, there are too many tables presented without texts to explain each table.

6. PLOS authors have the option to publish the peer review history of their article (what does this mean?). If published, this will include your full peer review and any attached files.

**Do you want your identity to be public for this peer review?** For information about this choice, including consent withdrawal, please see our Privacy Policy.

Reviewer #1: No

Reviewer #2: **Yes: **Jacob Novignon

---

## [Editor Report · Decision Letter 1]

24 Nov 2021

Tax abuse - the potential for the Sustainable Development Goals

PGPH-D-21-00278R1

Dear Dr. Lopez

We're pleased to inform you that your manuscript has been judged scientifically suitable for publication and will be formally accepted for publication once it meets all outstanding technical requirements.

Within one week, you'll receive an e-mail detailing the required amendments. When these have been addressed, you'll receive a formal acceptance letter and your manuscript will be scheduled for publication.

An invoice for payment will follow shortly after the formal acceptance. To ensure an efficient process, please log into Editorial Manager at https://www.editorialmanager.com/pgph/ click the 'Update My Information' link at the top of the page, and double check that your user information is up-to-date. If you have any billing related questions, please contact our Author Billing department directly at authorbilling@plos.org.

Kind regards,

Augustine D. Asante

Academic Editor

Additional Editor Comments (optional):

Please address the following minor comments:

Lines 48-50: "The SDG 4 and 6 targets are to ensure that all children complete free primary and secondary education and have access to safe water and safe sanitation by 2030" -- Please change the safe water to safe drinking water.

Lines 61-62: "There has been significant progress towards SDG 6, with two billion people gaining access to safe drinking water and 2.4 billion to safe sanitation between 2000 and 2020". - Please provide a reference for this sentence.

Lines 67: "88% of all children complete secondary education in North America and Europe, compared to only 29% in Sub-Saharan Africa, where only 62% of children complete primary education". Since you are beginning a new sentence, write the 88% in words.